# Blue Light and Temperature Actigraphy Measures Predicting Metabolic Health Are Linked to Melatonin Receptor Polymorphism

**DOI:** 10.3390/biology13010022

**Published:** 2023-12-30

**Authors:** Denis Gubin, Konstantin Danilenko, Oliver Stefani, Sergey Kolomeichuk, Alexander Markov, Ivan Petrov, Kirill Voronin, Marina Mezhakova, Mikhail Borisenkov, Aislu Shigabaeva, Natalya Yuzhakova, Svetlana Lobkina, Dietmar Weinert, Germaine Cornelissen

**Affiliations:** 1Department of Biology, Tyumen Medical University, 625023 Tyumen, Russia; 2Laboratory for Chronobiology and Chronomedicine, Research Institute of Biomedicine and Biomedical Technologies, Tyumen Medical University, 625023 Tyumen, Russia; kvdani@mail.ru (K.D.); h_aislu@mail.ru (A.S.); 3Tyumen Cardiology Research Center, Tomsk National Research Medical Center, Russian Academy of Sciences, 634009 Tomsk, Russia; 4Institute of Neurosciences and Medicine, 630117 Novosibirsk, Russia; 5Department Engineering and Architecture, Institute of Building Technology and Energy, Lucerne University of Applied Sciences and Arts, 6048 Horw, Switzerland; oliver.stefani@hslu.ch; 6Laboratory for Genomics, Proteomics, and Metabolomics, Research Institute of Biomedicine and Biomedical Technologies, Medical University, 625023 Tyumen, Russia; sergey_kolomeychuk@rambler.ru (S.K.); alexdoktor@inbox.ru (A.M.); voroninka@tyumsmu.ru (K.V.); yuzhakova@tyumsmu.ru (N.Y.); 7Laboratory of Genetics, Institute of Biology of the Karelian Science Center, Russian Academy of Sciences, 185910 Petrozavodsk, Russia; 8Department of Biological & Medical Physics UNESCO, Medical University, 625023 Tyumen, Russia; 9Department of Molecular Immunology and Biotechnology, Institute of Physiology of the Federal Research Centre Komi Science Centre, Ural Branch of the Russian Academy of Sciences, 167982 Syktyvkar, Russia; borisenkov@physiol.komisc.ru; 10Healthcare Institution of Yamalo-Nenets Autonomous Okrug “Tarko-Sale Central District Hospital”, 629850 Urengoy, Russia; dr.lobkina@mail.ru; 11Institute of Biology/Zoology, Martin Luther University, 06108 Halle-Wittenberg, Germany; dietmar.weinert@zoologie.uni-halle.de; 12Department of Integrated Biology and Physiology, University of Minnesota, Minneapolis, MN 55455, USA; corne001@umn.edu

**Keywords:** light, Arctic, actigraphy, body mass index, temperature, rs10830963, light at night, leptin, cortisol, metabolism

## Abstract

**Simple Summary:**

We examined the relationship between the circadian rhythms of activity, wrist temperature (wT), blue light exposure (BLE) and sleep with proxies of metabolic health (body mass index (BMI), leptin and cortisol) in Arctic residents during a single week at the spring equinox. We found that nocturnal BLE within a distinct time window is associated with a higher BMI but not with the actigraphy indices of sleep or motor activity. We determined an evening time window of non-overlapping 95% confidence intervals of BLE between groups with BMI < 25 and BMI > 25 kg/m^2^ between 9:30 p.m. and 0:30 a.m., with a threshold that is less than 1 lx. The area under the curve delineated by nocturnal BLE above recommended reference values represents the novel nocturnal excess index (NEI*bl*), which was the best actigraphy-based predictor of BMI. A higher BMI was also associated with lower 24 h wT, an association mostly characterizing those carrying the MTNR1B rs10830963 G-allele. In this allele carriers, a higher BMI was also closely related to NEI*bl*. A higher wake-after-sleep onset (WASO) correlated with higher leptin. Higher cortisol was associated with another novel index, the Daylight Deficit Index of blue light, DDI*bl*, and with an earlier onset of BLE.

**Abstract:**

This study explores the relationship between the light features of the Arctic spring equinox and circadian rhythms, sleep and metabolic health. Residents (N = 62) provided week-long actigraphy measures, including light exposure, which were related to body mass index (BMI), leptin and cortisol. Lower wrist temperature (wT) and higher evening blue light exposure (BLE), expressed as a novel index, the nocturnal excess index (NEI*bl*), were the most sensitive actigraphy measures associated with BMI. A higher BMI was linked to nocturnal BLE within distinct time windows. These associations were present specifically in carriers of the MTNR1B rs10830963 G-allele. A larger wake-after-sleep onset (WASO), smaller 24 h amplitude and earlier phase of the activity rhythm were associated with higher leptin. Higher cortisol was associated with an earlier M10 onset of BLE and with our other novel index, the Daylight Deficit Index of blue light, DDI*bl*. We also found sex-, age- and population-dependent differences in the parametric and non-parametric indices of BLE, wT and physical activity, while there were no differences in any sleep characteristics. Overall, this study determined sensitive actigraphy markers of light exposure and wT predictive of metabolic health and showed that these markers are linked to melatonin receptor polymorphism.

## 1. Introduction

We dedicate this paper to the memory of our dear colleague, Dr. Konstantin Danilenko, who helped initiate this project but sadly passed away on 18 January 2023.

Light is a pivotal factor in maintaining the alignment of circadian clocks with environmental time cues and adjusting to natural seasonal changes [1,2,3]. The 24 h light–dark cycle provides a major signal strengthening the internal synchronization of circadian processes during consecutive cycles, increasing the circadian amplitude of physiological processes [4,5,6]. The benefits of light for the circadian system and health imperatively depend on the circadian phase and change from the early morning to the late evening hours [1,2,7]. At certain moments, the benefits of daylight fade, and the hazardous effects of light at night (LAN) emerge [7]. This turning point can depend on endogenous factors such as the intrinsic circadian period and habitual sleep phase. It also may depend on chronotype and the availability of sunlight and its timing which varies greatly over the year and most drastically with the seasons at high latitudes. Of note, outdoor light at night correlates with eveningness in adolescents [8], while sufficient daylight during camping advances the circadian phase [9,10].

The Arctic light environment is challenging to the circadian system, health and well-being [11,12,13,14,15,16]. Previous studies also suggested that disruption of the circadian system has an adverse impact on metabolic health [17,18,19,20,21,22,23,24,25,26].

Novel actigraphy wearables [27] allow the collection of objective, continuous 24/7 information not only on circadian and sleep markers, such as motor activity and wrist temperature, but also on light exposure. Some devices include distinct spectral domains [28] of which blue light is of greater interest since intrinsically photosensitive retinal ganglion cells (ipRGCs), also known as melanopsin-expressing retinal ganglion cells (mRGCs), are particularly sensitive to blue light [29,30]. After over twenty years of epidemiological studies, including actigraphy, evidence for an association between body mass index (BMI) and sleep parameters remains elusive [31]. On the other hand, light exposure in terms of its intensity and timing showed a correlation with body weight in adults [32]. Constantino et al. [33] reported an association between more fragmented rhythms of activity and light at night with higher BMI. This study also showed associations between higher urbanization and smaller 24 h amplitude of light exposure with an increased risk of being overweight or obese. In a community-dwelling adult population, a higher BMI was best associated with a smaller relative amplitude (RA) of the rest–activity pattern, an integrative index reflecting higher night activity to lower daytime activity in relation to higher BMI [34]. Similar results showing lower metabolic risks with larger RA were obtained in adolescents [35,36]. Such changes can be modulated by different patterns of light exposure, as physical activity and light exposure are closely related [37]. Indeed, Obayashi et al. found that mean exposure to nighttime light above 3 lux is associated with a higher BMI and waist-to-height ratio [38] and a higher risk of diabetes [39]. Recent work in 552 community-dwelling adults, 63–84 years of age, linked LAN to actigraphic changes—delayed onset of least nocturnal activity and light exposure (as measured by the L5 index: 5 h of least activity/light exposure), lower inter-daily stability and smaller amplitude of activity and light exposure, which were associated with the prevalence of obesity, diabetes and hypertension [40].

Actigraphic studies incorporating the analysis of light exposure in association with BMI or other proxies of metabolic health in Arctic residents are currently lacking. The physiological and circadian effect of light depends on the timing and duration of light exposure. Therefore, we hypothesized that there is a need to introduce a novel relevant index for the assessment of the cumulative effect of nocturnal light excess and to compare it with other actigraphy-derived measures. Our study was performed in nearby locations in the Arctic; all data were collected during the same week within each season.

We included the rs10830963 polymorphism of the melatonin receptor MTNR1B gene since it is associated with metabolic risks and circadian rhythms [41,42,43,44,45,46] and may serve as a connection between circadian disruption due to impaired light signaling and metabolic response [41,46], including the circadian rhythm of temperature [44]. The pineal gland hormone melatonin is a well-documented substance that mediates the influence of outdoor light cues on the inner circadian machinery. Seasonal changes in the length of the dark phase lead to parallel shifts in the duration and timing of the onset of melatonin secretion. Melatonin exerts its effects via its own receptors, MTNR1A and MTNR1B, as well as through receptor-independent mechanisms [15]. Polymorphic variations in the MTNR1B gene have a profound effect on gene expression and protein properties. Associations between the carriage of the MTNR1B rs10830963 minor G-allele and the affected modulation of the metabolic response to melatonin and light were evident in previous studies [42,43,44]. Carriers of the minor G-allele of the MTNR1B rs10830963 snp represent a large part of populations that have elevated fasting glucose and higher risks of metabolic disorders, particularly type 2 diabetes mellitus [41]. They also have changed and prolonged patterns of melatonin secretion [42] that can be related to chronotype [43] or reduced light signaling due to retinal ganglion cell loss [44]. Also, carriers of this polymorphism can gain different effects as an outcome of weight loss programs [47].

## 2. Materials and Methods

This cross-sectional “Light Arctic” study adhered to the tenets of the Declaration of Helsinki and was approved by the Ethics Committee of Tyumen State Medical University (Protocol No. 101, 13 September 2021). Written informed consent was obtained from all participants.

### 2.1. Subjects and Data Collection

The “Light Arctic” study, phase 1 spring equinox, was conducted on residential populations of three nearby locations: Salekhard, 66°53′ Latitude North, 66°60′ Longitude East, Aksarka settlement, 66°33′ Latitude North, 67°48′ Longitude East, during 22 March–8 April 2022, and Urengoy town, 65°58′ Latitude North, 76°63′ Longitude East, during 18 March–2 April 2023. Dates were chosen to be as close to the equinox as possible and to avoid covering dates of yearly local celebratory community events. Voluntary participants (N = 156) aged 12 to 59 years were enrolled; 62 (38 representing non-native and 24 native populations) completed the study, providing data of sufficient quality according to the study protocol. All participants (mean age ± SD: 36.24 ± 14.21 years; 82.3% women) had a 5-day work/2-day free weekly schedule; none was engaged in night shift work, crossing more than 2 time zones 3 weeks before the survey or diagnosed with a sleep disorder. Exclusion criteria were severe and/or unstable diseases, including neurodegenerative diseases, history of acute coronary or cerebral blood flow disorders, severe cardiac arrhythmias, arterial hypertension of the third stage or uncontrolled blood pressure, angina pectoris FC III-IV, mental illness, drug and/or alcohol addiction, type I diabetes mellitus, severe thyroid diseases, malignant neoplasms, epilepsy, tuberculosis, taking beta-blockers, acute illness during the last week before examination, pregnancy or breast feeding and children under 3 years of age. All adult participants were office workers and their offspring; 10 participants under 16 years represented residents of the Aksarka Boarding School for natives. Weight (in kilograms) and height (in meters) of study participants were measured using medical height meter and scales at local medical units where participants arrived on the last day of actigraphy (the day of blood sampling). They were used to calculate body mass index (BMI = weight/height^2^). BMI of 12- to 16-year-olds was adjusted for age and sex for categorization into normal weight/overweight/obesity, as previously recommended [48].

### 2.2. Actigraphy

All participants completed sleep diaries and provided 7-day actigraphy wearing the ActTrust 2 (Condor Instruments, SP, Brazil) hardware on the non-dominant hand during the second week of the study. Motor activity (Proportional Integrative Mode, PIM), wrist skin temperature (wT), light intensity in lux (LI) and blue light intensity (BLI) in μw/cm^2^ were measured using ActTrust 2 at 1 min intervals. PIM estimates the area under the rectified analog signal for each epoch and measures movement intensity by summing the deviations from 0 V every 10th of a second [49]. PIM was used since it was shown to provide the highest correlation with polysomnography for sleep characteristics [50]. BLI measured with the RGB (Red Green Blue) sensor of ActTrust 2 has an out-of-range sensitivity with a secondary sensitivity peak at a wavelength of 680 nm [51] that comprises the match to the melanopic sensitivity curve to an otherwise close match between the B sensor (peak sensitivity at 470 nm) and the sensitivity of melanopsin. The mismatch between the B sensor and the melanopic sensitivity curve can be expected to be around 25% [52]. Of note, the spectral sensitivity of non-visual responses might change over time [53]. Parametric indices (MESOR, 24 h amplitude and acrophase) were estimated for PIM, wT and exposure to ambient light and blue light, and non-parametric indices inter-daily stability (IS), intra-daily variability (IV), relative amplitude (RA), circadian function index (CFI), the maximum 10 h period (M10), M10 onset, the least 5 h period (L5) and L5 onset were estimated for PIM and blue light intensity using the ActStudio software (Condor Instruments, São Paulo, Brazil) (https://condorinst.com/en/actstudio-software/ accessed on 15 May 2023). In addition, we calculated our novel indices: blue light Daytime Deficit Index (DDI*_bl_*), the area of blue light exposure below the reference curve between 6 a.m. and 8 p.m., and blue light nocturnal excess index (NEI*_bl_*), the area of blue light exposure above the reference curve between 8 p.m. and 5 a.m. Reference curves (Appendix A) considered close to natural gradual changes in blue light exposure were based in part on recent blue light exposure recommendations [7]. The international standard CIE S 026:2018 “CIE System for Metrology of Optical Radiation for ipRGC-Influenced Responses to Light”, defines five spectral weighting functions for the five retinal photoreceptor classes (S-cones, M-cones, L-cones, rods and melanopsin-based photoreception of ipRGCs) [54]. The five α-opic irradiances are expressed in μw/cm^2^, and the corresponding photometric quantities are the five α-opic equivalent daylight illuminances (“α-opic EDIs”), which are expressed in lux (CIE. S 026/E:2018.). Each α-opic EDI represents the (equivalent) illuminance of the standard illuminant D65 that produces the same α-opic irradiance as the test light. Brown et al. recommend at least 250 lx melanopic EDI (mEDI) (33 μw/cm^2^) during daytime, no more than 10 lx mEDI (1.33 μw/cm^2^) three hours before bedtime and less than 1 lx mEDI (0.1 μw/cm^2^) during sleep. Further details for calculations of DDI*_bl_* and NEI*_bl_* are provided in Appendix A.

The following sleep parameters were also calculated using the ActStudio software: bedtime, wake time, time in bed, total sleep time, sleep efficiency, sleep latency and wake after sleep onset (WASO), or minutes awake during the sleep period between sleep time and wake time. Sleep diaries were used to check the data and remove artifacts. Volunteers were also asked to trigger an alarm button at times of going to bed and waking up. Primary analyses of sleep–wake data were performed using the ActStudio software, which applies algorithms based on the method of Cole et al., 1992 [55].

### 2.3. Biochemical Assessment

After completion of actimetry, all participants provided blood samples, which were collected using a vacutainer from the ulnar vein, in the morning (8:00–9:00), after a 12 h fast. Blood samples were washed with a HydroFlex microplate mixer, Tecan, Grödig, Austria. The biochemical kit Leptin Sandwich ELISA produced by DRG Instruments GmbH (Marburg, Germany) and a set of reagents for quantitative enzyme immunoassay of cortisol in human serum (SteroidIFA-cortisol, St. Petersburg LLC Alkor Bio, St. Petersburg, Russia) were used. All biochemical tests were performed in the certified laboratory of the Tyumen Medical University’s Research Institute of Biomedicine and Biomedical Technologies.

### 2.4. Genotyping

The same operator, who was not aware of the participant’s characteristics, performed the genotyping. Blood samples were collected via standard protocols from 50 participants. DNA was isolated from patients’ samples using DiaGene DNA Isolation Kit (Dia-M; Moscow, Russia) according to the manufacturer’s instructions. The real-time polymerase chain reaction was performed using iCycler Real Time System with iQ5 Manager software of Bio-Rad Laboratories Inc. (Hercules, CA, USA). Polymorphic gene variants were identified using SNP Screen Kit (Syntol; Moscow, Russia) for MTNR1B rs10830963. In each reaction, two allele-specific hybridizations were used to detect two alleles of the studied polymorphism, independently on two fluorescence channels (ROX and FAM).

### 2.5. Data and Statistical Analyses

In addition to ActStudio, which provided primary analyses of actigraphy data, original files were used to analyze raw, normalized and log_10_-transformed data of ambient light exposure, blue light exposure, PIM and wT by ANOVA and cosinor analysis [56]. The software packages Excel, STATISTICA 6 and SPSS 23.0 were used for statistical analyses. ANOVA/MANOVA tests of equality of group means were performed. The Shapiro–Wilk’s W-test was applied to check for normality of distributions. If variables were distributed normally (W-test’s *p*-value > 0.05), a one-way ANOVA was used. Otherwise, the Mann–Whitney, Kruskal–Wallis and post hoc tests were used. A statistical comparison of correlation strength was performed with the cocor free online software [57]. Linear regression analyses were applied to assess the relationship of actigraphy measures with BMI, leptin and cortisol. Multiple regression analyses with co-factors of sex and population (natives vs. non-natives) and age as continuous variable were added to regression models. The level of statistical significance was set at 5%. Benjamini–Hochberg’s False Discovery Rate correction using an FDR value 0.1 or higher was applied to adjust *p*-values for multiple testing. R charts were constructed to describe linear associations of light exposure at a given time epoch with BMI and leptin values. Correlation coefficients from linear regression of 48 consecutive 30 min averages of BLE obtained from each participant were used for ANOVA and cosinor analyses to describe the dependence of these r values on time.

## 3. Results

### 3.1. Seasonal Features of Light Exposure in the Arctic Spring as a Season with the Most Favorable Circadian Light Environment

Participants of the “Light Arctic” study provided seven-day actimetry in each season: during the winter solstice, spring equinox, summer solstice and autumn equinox. For this paper, spring was chosen as a reference point to provide seasonal assessments for associations of the ambient light environment with circadian physiology, sleep and metabolism. Figure 1 depicts 24 h patterns of light exposure during four Arctic seasons (around spring and autumn equinoxes, around polar summer and polar winter). Drastic changes in ambient light exposure are evident. Individual mean values of blue light exposure in consecutive 30 min epochs were used to depict the average 24 h patterns of light exposure in each season. Light exposure around the spring equinox was closest to the recommended reference curve for circadian light hygiene [7] (Figure 1). The lower row of Figure 1 depicts mean values of raw blue light exposure against the reference curve and the optimal light exposure log_10_-transformed values of blue light exposure for consecutive 30 min epochs.

### 3.2. General Characteristics of Participants of the Spring Equinox Session

Table 1 provides general characteristics of BMI and age depending on sex, indigeneity and MTNR1B rs10830963 polymorphism for all participants and separately within two age groups. There were no differences for mean age or BMI within each age group (12–16 y or >20 y), between sexes, populations or between carriers and non-carriers of the MTNR1B rs10830963 G-allele. Overall, natives have lower compliance with protocol requirements; 10 out of the 24 natives (41.7%) who completed the study represented the Aksarka Boarding School. Hence, natives’ mean age and BMI were lower overall. All participants of the younger age group were classified as normal-weight after correction for age and sex. They were considered together with lean adults (BMI < 25) in primary statistical analyses before corrections for age, sex and indigeneity were applied.

### 3.3. Blue Light Nocturnal Excess and Lower Wrist Temperature Predict Body Mass Index

BMI was best predicted by leptin, wrist temperature (wT) MESOR and our novel index, NEI*_bl_*. Table 2 shows actigraphy-based associations with BMI in the Arctic residents during the spring equinox. NEI*_bl_* and wT MESOR were validated as the best predictors for BMI either after Benjamini–Hochberg’s FDR correction at FDR = 0.1 or in a multiple regression model after correction for the co-factors of age, sex and population.

Multiple regression model and stepwise forward multiple regression analyses confirm that these two actigraphy-based endpoints improve the prediction of BMI after considering age, gender and population as co-factors. In a stepwise forward multiple regression model (r^2^ = 0.637, *p* < 0.0001), NEI*_bl_* and wT MESOR remained predictors of BMI: leptin (standardized coefficient b = 0.619; *p* < 0.0001), wT MESOR (b = −0.298; *p* = 0.001), NEI*_bl_* (b = 0.242; *p* = 0.006) and gender (b = −0.184; *p* = 0.051).

Figure 2 depicts particular hours when the association between blue light exposure and BMI is statistically significant. Not only are there differences in nocturnal blue light exposure among the three BMI groups, but the association (gauged by the correlation coefficient) also depends on time. The time course of r values between BMI and blue light exposure in successive 30 min epochs over the whole 24 h day can be approximated by a cosine function with an estimated acrophase of 23:34 (11:34 p.m.). Furthermore, we determined an evening time window of non-overlapping 95% confidence intervals of blue light exposure between groups of participants with a BMI <25 or >25, Appendix A.

### 3.4. MTNR1B Polymorphism Accounts for the Interaction between Light, Wrist Temperature and Metabolic Health

Associations between a lower wT MESOR and a higher BMI (Figure 3), as well as between higher nocturnal blue light exposure and a higher BMI (Figure 4), are particularly strong among carriers of the G-allele of MTNR1B rs10830963 C > G SNP polymorphism but not evident among those with the MTNR1B rs10830963 CC genotype. The correlation between BMI and NEI*_bl_* is statistically significant among the G-allele carriers (r = 0.465; *p* = 0.029) but not among those with the CC genotype (r = 0.104; *p* = 0.599). The Time * BMI * MTNR1B interaction for log_10_(blue light) is also significant (F_(47, 2832)_ = 1.504, *p* = 0.015) (Figure 4), as is the BMI * MTNR1B interaction for NEI*bl* (F_(1, 46)_ = 5.083, *p* = 0.029); NEI*_bl_* is significantly higher in MTNR1B rs10830963 G-allele carriers with a BMI > 25 (*p* = 0.031). There are no other significant differences for actigraphy-based characteristics between carriers and non-carriers of the MTNR1B G-allele depending on BMI (Appendix A).

### 3.5. Associations of Leptin and Cortisol with Actimetry-Derived Indices

Table 3 contains details of actigraphy-derived associations with leptin and cortisol in Arctic residents during the spring equinox. After correction for multiple testing at Benjamini–Hochberg’s FDI = 0.1, a higher wake-after-sleep onset (WASO) and smaller M10, MESOR and 24 h amplitude of activity were associated with higher leptin (r = −0.339, *p* = 0.008, and r = −0.289, *p* = 0.024, respectively); rather surprisingly, higher leptin was also associated with an earlier 24 h phase of activity (r = −0.279, *p* = 0.030) and lower wrist temperature (r = −0.302, *p* = 0.014). In a multiple regression model, after correction for co-factors of age, sex and population, only the correlation with WASO remained significant (b = 0.240; *p* = 0.049). An earlier M10 onset of blue light exposure and a higher Daylight Deficit Index, DDI*_bl_*, were associated with higher cortisol (r = −0.317, *p* = 0.014, and r = 0.254, *p* = 0.050, respectively) but not after correction for multiple testing with Benjamini–Hochberg’s FDI = 0.1. An earlier M10 onset of BLE and DDI*_bl_*, however, remained equally significant correlates of higher cortisol in a multiple regression model after correction for the co-factors of age, sex and population (whole model r^2^ = 0.255; DDI*_bl_*, b = 0.353; *p* = 0.011; M10 onset, b = 0.310; *p* = 0.016).

### 3.6. Age-, Sex- and Indigeneity-Related Aspects of Actigraphy-Based Indices

Natives had a significantly higher DDI*_bl_* (*p* < 0.05) and lower NEI*_bl_* (*p* < 0.001), despite the fact that other parametric and non-parametric indices of blue light exposure did not significantly differ from those of non-natives (Table 4). Natives had higher overall daytime activity, MESOR, 24 h amplitude, M10 (*p* < 0.001) and wT MESOR (*p* < 0.01), suggesting that between-group differences in blue light indices were not related to less outdoor light exposure that is commonly linked to motor activity. Adult participants had an earlier 24 h phase of blue light exposure and a lower DDI*_bl_* (*p* < 0.01) that led to a higher RA*bl* (*p* < 0.05), despite having higher nocturnal blue light exposure, as gauged by NEI*_bl_* (*p* < 0.01) or log_10_(L5) (*p* < 0.01). Adults also had a higher and later onset of L5 activity (*p* < 0.05), a smaller circadian amplitude (*p* < 0.05) and a smaller RA of activity (*p* < 0.001). Male participants had more blue light exposure, gauged by the MESOR, 24 h amplitude (*p* < 0.01) or M10 (*p* < 0.05). They, however, had a lower IS and a later 24 h phase of activity (*p* < 0.05). There were no other significant differences in actigraphy-derived sleep characteristics between populations, sexes or age groups.

## 4. Discussion

Arctic residents are exposed to ambient light with drastic seasonal differences. The amount of light and the alternation and duration of light/dark phases within 24 h change swiftly during spring and autumn. It reaches marginal conditions such as polar day (sun remains above the horizon throughout the 24 h) and polar night (sun’s disc is visible above the horizon for less than three hours for 30 days). At 66- to 67-degree latitude North (locations of this study), polar summer days last for 30 days (7 June—6 July), and the day length in the winter is less than three hours for about 4 weeks (8 December—4 January). We compared real-time light exposure in this longitudinal year-long survey in Arctic residents among week-long records obtained within each season. We observed that the spring equinox was the most comfortable season in terms of dynamic light features, which fit most closely the recommendations for 24 h indoor light schedules [7].

We started our analyses of actigraphy data and their relationship with metabolic proxies of health and genetic factors during the spring equinox. We found that a higher BMI is linked to a lower wT MESOR and a higher evening BLE. The association between BMI and BLE occurs exclusively within distinct time windows and is best expressed by an area above the reference curve, represented by a sine function fitted over the recommended threshold for the proper light hygiene standards (Figure 1). Other actigraphy indices of sleep or motor activity were not closely linked to BMI. Remarkably, associations of BMI with wrist temperature and evening BLE are closely coupled to carrying the MTNR1B rs10830963 G-allele. Importantly, these differences in evening BLE were subtle, however clearly pronounced after log_10_ transformation of BLE data, and were linked to a particular time window.

The timing of light exposure is critical [58,59] and discriminates between its beneficial and harmful effects [21,60]. The 24 h amplitude of light signaling, or the dynamic range between daytime and nighttime light exposure, is another factor that is decisive for health and well-being [21,26,61,62]. Overall, daylight is beneficial and strengthens the alignment power of light signaling [4,9,63], whereas light at night is linked to circadian misalignment and is detrimental to health [64,65,66]. Light at night is linked to adverse metabolic changes and weight gain in laboratory animals [67,68,69] and humans [60,70,71,72,73,74].

Morning bright light treatment reduced body fat and appetite in overweight women [75]. Light exposure above a 500 lux threshold early during the daytime is associated with lower body weight independently of sleep timing, duration and total activity in adults [32]. The direction of this relationship, however, was opposite in children [76]. Individual differences in metabolic responses to light and circadian misalignment by light at the wrong time are pronounced [77] and depend on light sensitivity that can be modulated by genetic factors [78] and sex [79,80] and reduced with advancing age [81,82]. Light sensitivity can differ within a 50-fold range among individuals. Light sensitivity may depend on yet unrecognized genetic factors distinct from those determining the chronotype, circadian phase and intrinsic period, as no relationship between individual sensitivity to light and dim light melatonin onset, DLMO, phase angle between DLMO and habitual bedtime or light-exposure history was found [83]. Some individuals are sensitive to very dim blue light [83,84], suggesting that even low blue light irradiance may have consequences for the circadian system and health. The effects of timing of light exposure on metabolic health throughout the 24 h scale could be best modeled by a sinusoidal function, as phase response curves to light exist for melatonin [85] and also for lipids and hepatic proteins [25].

As a gauge of peripheral thermoregulation, wrist temperature, which can be assessed by monitoring, is a promising digital biomarker for numerous diseases [86]. The relationship between the circadian rhythm of temperature and metabolic features is well known for mammalian species and humans. A higher body temperature and smaller 24 h amplitude are generally linked to a higher body mass [87]. The circadian dynamics of insulin resistance and glucose turnover are also associated with the circadian rhythm of temperature, affecting the relationship between heat production and heat loss during activity and rest periods [87,88,89]. A lower mean wT in obese vs. normal-weight women was found during three-day measurements with the Thermochron iButton [90], accompanied by a flattened 24 h wT pattern. It was associated with a dampened midday–midnight difference of cortisol and melatonin. A higher BMI was associated with a smaller amplitude and a lower intra-daily variability of the 24 h rhythm of wT in 192 adult women [91]. Similarly (since physiological and circadian relations between wT and body temperature are opposite), a higher average (“over the entire study”) core temperature in obesity and a higher core temperature in women vs. men were reported [92]. Also, a higher nocturnal body temperature in pre-diabetic patients and a dampened 24 h amplitude in diabetic patients were observed [93]. Another study, however, found sex-related (higher in women) but not obesity-related differences in core temperature [94], alerting us to the need to study mutations in core-temperature-regulating genes. Indeed, in this work, we discovered a close MTN1B-linked relationship for the association between the wT MESOR and BMI. This finding is in agreement with previous studies. MTNR1B also mediates the effect of melatonin on insulin gene expression [95]. The MTNR1B rs10830963 G-allele may be a link between circadian rhythm alterations, metabolic disorders and diabetes [41,96]. This SNP is associated with disrupted circadian phenotypes and also with altered melatonin secretion [42,43,44]. Both factors are prerequisites for metabolic disturbances and diabetes [41].

At night and during human sleep, heat production decreases, and heat loss from distal body parts increases [88]. Light at night affects metabolism and energy expenditure. Under controlled conditions of constant activity, evening and nocturnal light affected body temperature [97,98,99], possibly by influencing melatonin secretion [100]. BLE prior to bedtime significantly decreased fat oxidation and increased the respiratory quotient, an indicator of the carbohydrate-to-fat oxidation ratio [101]. It may increase obesity risk due to disturbances of circadian rhythms affecting resting/nocturnal nutrient metabolism. Recent work showed that the ambient light environment modulates postprandial metabolism, thermo-modulation and energy expenditure in a time-of-day-dependent manner [59].

Herein, we show that BLE is subtly, though significantly, higher within a distinct time window in individuals with a higher BMI. This association is linked to the presence of the MTNR1B G-allele. Remarkably, these distinct features of nocturnal BLE are not linked to any differences in actigraphy-derived characteristics of sleep or motor activity but are linked to a lower wT MESOR. Overall, this result prompts an assumption that, at least in Arctic residents, enhanced nocturnal light hygiene to prevent blue light overexposure above the threshold (Appendix A) could be most effective in carriers of the G-allele of the MTNR1B rs10830963 C > G SNP polymorphism to support metabolic health. Of note, this threshold relates to values of light exposure that are lower than 1 lx.

We also found an association between a higher WASO and higher leptin and higher cortisol with a higher DDI*_bl_* and with an earlier M10 onset of BLE that remained statistically significant after accounting for age, sex and indigeneity. Leptin and cortisol are stress-marker hormones, closely linked to metabolic health [102]. On the other hand, these hormones have pronounced circadian rhythms that are somewhat inverse in phase, suggesting antagonistic modulation, at least in some individuals [102]. Leptin inhibits the action of cortisol centrally in the hypothalamus [103] and peripherally in the adrenal gland [104]. Leptin increases with a higher BMI and usually is higher in women than men [105,106]. Leptin has an evident 24 h rhythm with an average amplitude above 20% of the mean and a nocturnal phase similar to that of melatonin [107]. Unlike melatonin, the amplitude and phase of the 24 h rhythm in leptin seem to not be affected by the metabolic state [107]. Conditions of compromised sleep exposure to morning blue light increased leptin [108], while nighttime use of electronic devices resulted in moderate suppression of leptin [109]. However, no effect of blue-enriched light or timing of its exposure (morning vs. evening) was found in conditions of non-deprived sleep [58]. Bright light of about 10,000 lux reduces cortisol at both rising and descending phases when compared to dim light conditions [110]. Daytime bright light, however, had no effect [111]; there was no effect either of dynamic “natural light modulation” scenarios on cortisol [112]. These findings suggest that the effect of light on cortisol may depend on previous light exposure and on the extent of difference between introduced light exposure vs. preexisted light exposure.

The current study has its strengths and limitations. The major strength of the study was that all actigraphy was performed simultaneously during the same week of the same season in nearby geographical locations. Limitations include the fact that most volunteers in this study were women; results herein hence apply mainly to adult women residing in the Arctic. Results may also be biased by individual differences related to the menstrual cycle. Therefore, this aspect should be addressed in future studies.

## 5. Conclusions

This study revealed that in Arctic residents during the spring equinox, a higher BMI is linked to nocturnal BLE within distinct time windows and to a lower wT MESOR, irrespective of any detectable differences in the actigraphy indices of sleep or motor activity. This link is coupled to the MTNR1B rs10830963 G-allele. We determined actigraphy-based predictors of leptin and cortisol. We also provided low nocturnal BLE thresholds for optimal metabolic health between a 9 p.m. and 1 a.m. time window and suggested a novel index, which estimates such excess.

In our future work, we will examine follow-ups of these findings during other seasons, particularly during winter as a most light-deficient season in the Arctic.

## 6. Patents

Three patents (RU) are pending.

## Figures and Tables

**Figure 1 biology-13-00022-f001:**
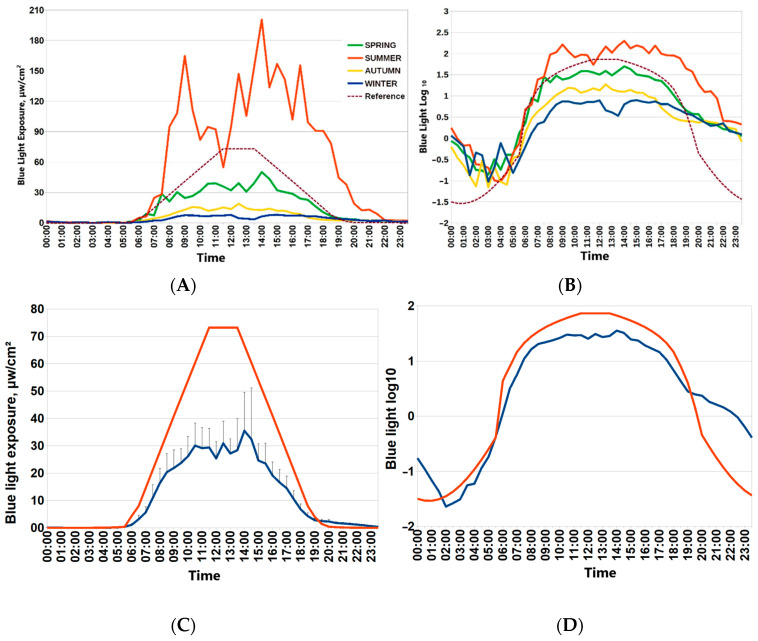
Twenty-four-hour patterns of blue light exposure in the Arctic by seasons. (**A**,**B**) Mean blue light exposure during four seasons (spring equinox, autumn equinox, winter polar night, summer polar day) in Arctic residents. (**C**) Mean values of blue light exposure for consecutive 30 min epochs during spring equinox (blue line) against reference curve of optimal light exposure (red line). Vertical bars denote 95% confidence intervals. Reference curve for optimal blue light exposure is dashed brown line. (**D**) log_10_-transformed values of blue light exposure for consecutive 30 min epochs during spring equinox (blue line) against log_10_-transformed reference curve of the optimal light exposure (red line).

**Figure 2 biology-13-00022-f002:**
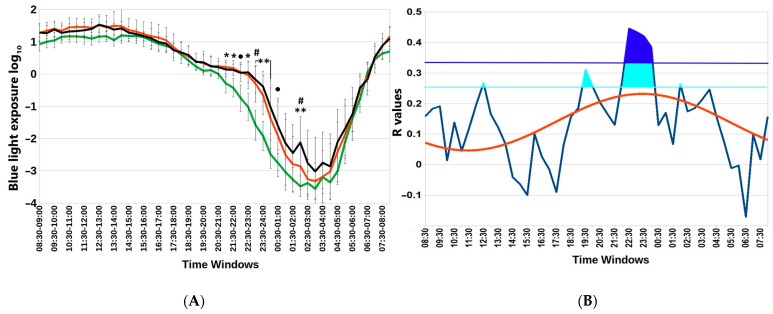
Subtle differences in nocturnal blue light exposure challenge metabolic health in the Arctic spring. (**A**) Distinct 30 min time windows of blue light exposure in Arctic residents with different body mass index (BMI) values. Twenty-four-hour patterns of blue light exposure, expressed as log_10_ in normal weight (BMI < 25); overweight (BMI = 25–30) and obese (BMI > 30) individuals are depicted. Vertical bars denote 95% confidence intervals. Green lines BMI <25; red lines BMI >25; black lines BMI >30 ● *p* < 0.1 (22:00–22:30; 00:30–01:00); * *p* < 0.05 (21:00–22:00); ** *p* < 0.01 (23:00–00:30; 02:00–02:30) for mean-rank differences between BMI < 25 and 25–30; # *p* < 0.05 (23:00–23:30 and 02:00–02:30) for mean-rank differences between BMI = 25–30 and >30. Note higher exposure to blue light in the evening and early night among overweight and obese compared to lean participants. (**B**) Chart of r-values from linear regression of BMI with respect to blue light exposure (ordinate) in consecutive 30 min time windows (abscissa) (blue curve). Horizontal light blue line corresponds to the threshold of significance at *p* < 0.05; dark-blue horizontal line: threshold of significance after Benjamini–Hochberg’s correction for multiple testing at 0.1. Red curve: cosinor model rejects zero-amplitude assumption of no rhythmicity (*p* = 0.002), acrophase = 23:24.

**Figure 3 biology-13-00022-f003:**
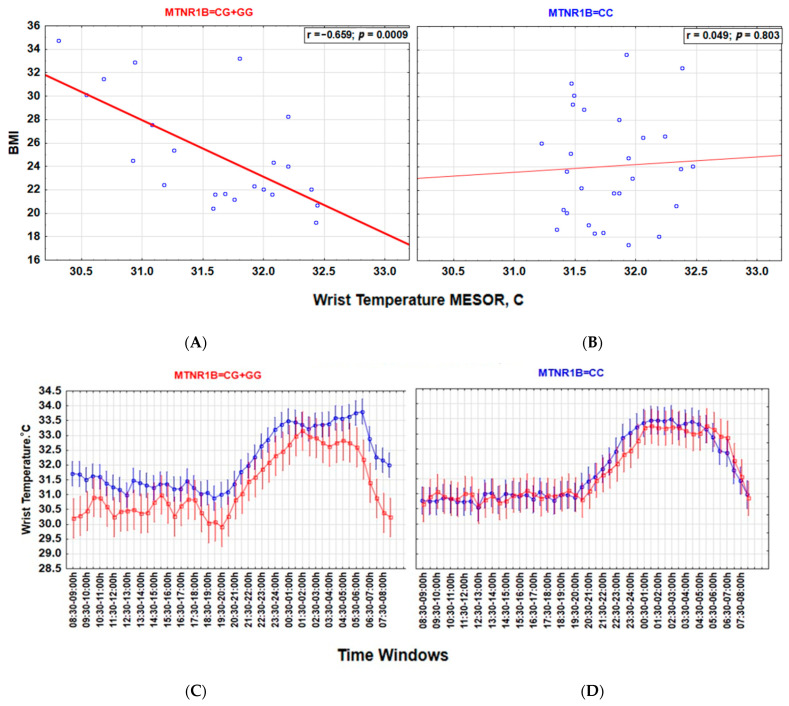
MTNR1B rs10830963 G-allele defines the association between body mass index (BMI) and wrist temperature (wT) MESOR. (**A**,**B**) Lower wT MESOR is associated with BMI in G-allele carriers but not in those with the CC genotype. Strength of correlation between wT MESOR and BMI is significantly stronger in MTNR1B G-allele carriers ((**A**), r = −0.659; *p* = 0.0009, n = 22) than in those with the CC genotype ((**B**), r = 0.049; *p* = 0.803, n = 28); z = −2.825, *p* = 0.005. (**C**,**D**) Twenty-four-hour patterns show lower wT, particularly in the morning, in overweight G-allele carriers than in normal-weight G-allele carriers (**C**); this difference is absent in those with the CC genotype (**D**). Vertical bars denote 95% confidence intervals. blue BMI < 25; red BMI > 25.

**Figure 4 biology-13-00022-f004:**
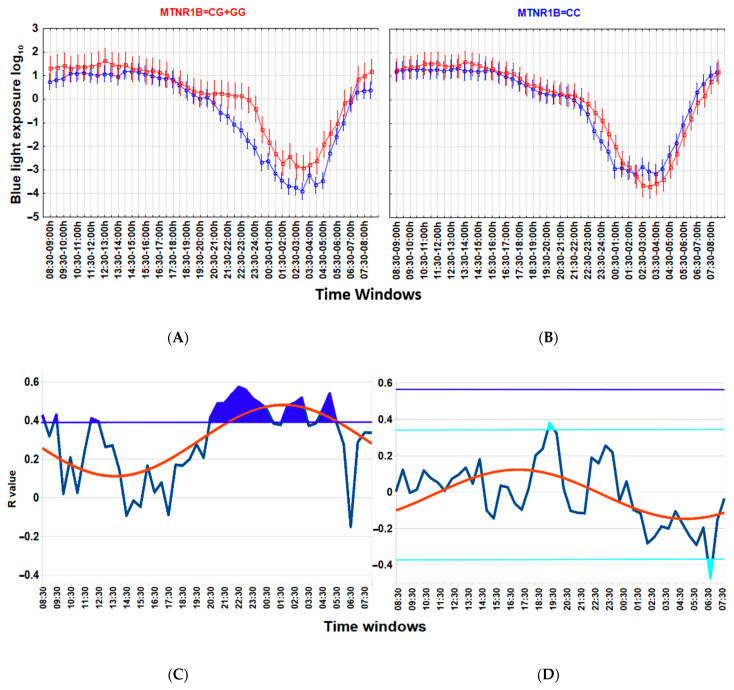
Association between higher nocturnal blue light exposure and higher body mass index (BMI) is linked to the MTNR1B rs10830963 G-allele. (**A**,**B**) ANOVA for Time*group interaction, blue light exposure log_10_ in MTNR1B G-allele carriers (**A**), F_(47, 1584)_ = 1.982, *p* = 0.0001; in those with the MTNR1B CC genotype (**B**), F_(47, 1248)_ =1.383, *p* = 0.046. Time * BMI * MTNR1B interaction for blue light exposure log_10_ is significant, F_(47, 2832)_ =1.504, *p* = 0.015. Thirty-minute time windows of normalized wrist temperature and activity in Arctic residents with different BMIs. Vertical bars denote 95% confidence intervals. (**C**,**D**) Chart of r-values from a linear regression of BMI with blue light exposure (ordinate) in consecutive 30 min time windows (abscissa) (blue curve). (**C**) G-allele carriers; (**D**) CC genotype (non-carriers). Horizontal light blue line corresponds to the threshold of significance at *p* < 0.05; dark-blue horizontal line corresponds to the threshold of significance after Benjamini–Hochberg’s correction for multiple testing at 0.1. Thresholds are similar for G-allele carriers. Red curve: cosinor model rejects zero-amplitude assumption of no rhythmicity; *p* < 0.00001, acrophase = 01:41 for G-allele carriers, CG + GG genotypes; *p* = 0.0002; acrophase = 17:15 for CC genotype. Correlation of the higher evening blue light exposure with the higher BMI after correction for multiple testing is significant in G-allele carriers during time epochs shaded in blue.

**Table 1 biology-13-00022-t001:** General characteristics of body mass index and age depending on sex, indigeneity and MTNR1B rs10830963 polymorphism and within age groups.

	Sex			Indigeneity			MTNR1B Genotype		
	Females (n)	Males (n)	*p*-Value	Non-Natives (n)	Natives (n)	*p*-Value	GG + GC (n)	CC (n)	*p*-Value
Age	28.4 ± 12.6 (51)	38.0 ± 14.0 (11)	0.041	40.7 ± 11.0 (38)	29.2 ± 15.9 (24)	0.001	39.8 ± 12.6 (22)	40.4 ± 11.4 (28)	0.854
Age, >20 y.	37.1 ± 3.8 (41)	43.7 ± 8.6 (7)	0.054	43.0 ± 8.0 (35)	42.0 ± 9.6 (13)	0.709	43.6 ± 8.4 (19)	42.5 ± 8.5 (26)	0.675
Age, 12–16 y.	13.0 ± 1.4 (10)	14.4 ± 1.4 (4)	0.109	14.0 ± 2.6 (3)	14.0 ± 1.2 (11)	0.999	15.3 ± 1.5 (3)	12.5 ± 0.7 (2)	0.100
BMI	24.2 ± 4.6 (51)	24.2 ± 4.9 (11)	0.995	25.4 ± 4.6 (38)	22.2 ± 4.6 (24)	0.010	25.1 ± 4.7 (22)	24.1 ± 4.7 (28)	0.456
BMI, >20 y.	25.1 ± 5.0 (41)	26.8 ± 3.4 (7)	0.489	25.8 ± 4.5 (35)	24.1 ± 5.4 (13)	0.278	25.8 ± 4.6 (19)	24.1 ± 5.4 (26)	0.312
BMI, 12–16 y.	20.2 ± 1.7 (10)	19.5 ± 1.5 (4)	0.389	20.3 ± 1.5 (3)	19.9 ± 1.7 (11)	0.750	20.4 ± 1.2 (3)	20.2 ± 2.2 (2)	0.896

Mean values and standard deviations are indicated. *p*-values are from one-way ANOVA. N in parentheses—number of participants included in the study.

**Table 2 biology-13-00022-t002:** Actigraphy-based associations with body mass index (BMI) in Arctic residents during spring equinox (n = 62).

Variable	BMI Groups				Regression with BMI		
	1. Normal (<25, n = 40)	2. Overweight (25–30, n = 11)	3. Obesity (30–35, n = 11)	K-W *p*-Value	r	*p*	MR + CF p
Activity, PIM ^#^							
MESOR	2606 ± 609	2226 ± 539	2708 ± 1075	0.257	−0.079	0.541	0.494
24 h A	2002 ± 638	1633 ± 447	1823 ± 853	0.182	−0.234	0.067	0.727
Phase	14:47 ± 1:16	14:54 ± 1:22	14:31 ± 1:16	0.854	−0.153	0.234	0.222
M10	4270 ± 1070	3650 ± 816	4288 ± 1861	0.208	−0.120	0.354	0.428
M10 Onset	9:08 ± 1:17	8:32 ± 1:25	8:58 ± 1:00	0.402	0.175	0.174	0.334
L5	231 ± 208	124 ± 69	271 ± 242	0.068	0.077	0.554	0.855
L5 Onset	1:34 ± 1:11	1:29 ± 1:13	2:04 ± 1:23	0.406	−0.123	0.342	0.843
IV	0.888 ± 0.167	0.847 ± 0.164	0.836 ± 0.180	0.484	−0.147	0.255	0.178
IS	0.563 ± 0.113	0.523 ± 0.103	0.497 ± 0.067	0.068	−0.186	0.147	0.894
RA	0.893 ± 0.065	0.903 ± 0.050	0.836 ± 0.071	0.056	**−0.345**	**0.006 ***	0.299
CFI	0.676 ± 0.058	0.663 ± 0.056	0.600 ± 0.071	0.546	−0.172	0.182	0.994
Wrist temperature, °C ^#^							
**MESOR**	**31.99 ± 0.50**	31.63 ± 0.40	**31.29 ± 0.64**	**0.002 ^1–3^**	**−0.491**	**<0.0001 ***	**0.008 ***
24 h A	1.42 ± 0.53	1.29 ± 0.57	1.29 ± 0.59	0.694	−0.072	0.576	0.161
Phase	2:58 ± 1:27	2:19 ± 1:43	2:22:1:50	0.397	**−0.322**	**0.011 ***	**0.012 ***
Sleep characteristics, hh:mm ^#^							
Bedtime	22:35 ± 0:56	22:59 ± 1:15	22:49 ± 1:33	0.587	−0.003	0.983	0.618
Wake time	6:51 ± 1:03	7:13 ± 1:38	7:07 ± 2:09	0.395	−0.028	0.826	0.576
Sleep phase	2:43 ± 0:54	3:06 ± 1:15	2:54 ± 1:33	0.514	−0.040	0.760	0.432
Time in bed	8:13 ± 0:50	8:13 ± 1:27	8:17 ± 2:06	0.663	0.013	0.921	0.931
Total sleep	7:19 ± 0:44	7:10 ± 1:18	7:22 ± 2:06	0.467	0.074	0.568	0.767
Sleep latency, min	02.05 ± 1.68	3.08 ± 2.92	3.18 ± 2.63	0.279	*−0.228*	*0.075*	0.174
Sleep efficiency, %	88.67 ± 4.03	85.58 ± 6.24	88.15 ± 6.73	0.396	−0.207	0.106	0.613
WASO	0:50 ± 0:21	0:58 ± 0:27	0:50 ± 0:30	0.619	0.147	0.254	0.721
Blue light, μw/cm^2^ ^#^							
MESOR	9.65 ± 7.00	15.23 ± 10.19	12.46 ± 5.75	*0.074*	0.158	0.221	0.146
24 h A	13.89 ± 9.78	22.54 ± 15.81	18.11 ± 9.57	0.136	0.149	0.249	0.167
Phase	12:58 ± 0:53	12:34 ± 0:51	12:58 ± 0:29	0.699	0.099	0.442	0.420
M10	19.81 ± 11.98	34.06 ± 23.28	26.44 ± 13.43	*0.075*	0.178	0.166	0.661
M10 Onset	7:57 ± 0:51	7:43 ± 0:39	7:57 ± 0:38	0.780	0.080	0.535	0.550
L5	0.06 ± 0.14	0.07 ± 0.16	0.10 ± 0.11	0.126	0.086	0.506	**0.006 ***
L5 Onset	1:15 ± 1:47	**0:40 ± 0:43**	**2:26 ± 2:20**	**0.035 ^2–3^**	−0.237	0.063	0.505
L5 log_10_	−2.42 ± 1.28	−2.29 ± 1.25	−1.55 ± 0.93	0.129	**0.308**	**0.015 ***	0.185
IV	0.937 ± 0.352	0.924 ± 0.361	0.879 ± 0.297	0.933	−0.041	0.752	0.168
IS	0.381 ± 0.133	0.444 ± 0.188	0.375 ± 0.105	0.455	0.103	0.428	0.110
RA	**0.988 ± 0.023**	0.985 ± 0.022	**0.969 ± 0.027**	**0.022 ^1–3^**	**−0.300**	**0.018 ***	0.148
DDI*_bl_*	367 ± 106	277 ± 150	313 ± 106	0.052	−0.185	0.150	0.943
**NEI*_bl_***	**1.49 ± 1.58**	**3.37 ± 3.38**	**3.39 ± 2.21**	**0.003 ^1–2/1–3^**	**0.417**	**0.0008 ***	**0.005 ***

All values in the groups are mean ± SD. 24 h A—24 h amplitude; M10—average value of 10 h of greatest activity or blue light exposure; L5—5 h of lowest activity or blue light exposure; IV—intra-daily variability; IS—inter-daily stability; RA—relative amplitude; DDI*_bl_*—daylight deficiency index; NEI*_bl_*—nocturnal excess index; CFI—circadian function index; K-W p—Kruskall–Wallis *p*-value, where numbers show significant differences between particular BMI groups; phases and onset times are indicated in hh:mm; DDI*_bl_* and NEI*_bl_* in μw/cm^2^ × hour; significant differences between the groups, post Tukey HST; and non-zero correlations are in **bold**. * Significant after correction for multiple testing at Benjamini–Hochberg’s FDR = 0.10; MR + CF p—p-value from multiple regression model including all related variables after correction for co-factors of age, sex and population. #—unless otherwise indicated.

**Table 3 biology-13-00022-t003:** Actigraphy-based associations with leptin and cortisol in Arctic residents during spring equinox (n = 62).

Variable	Leptin			Cortisol		
	r	*p*	MR + CF p	r	*p*	MR + CF p
Activity (power integrative mode)						
MESOR	**−0.288**	**0.024**	0.132	−0.174	0.184	0.374
24 h Amplitude	**−0.289**	**0.024 ***	0.583	−0.134	0.308	0.360
Phase	**−0.279**	**0.030 ***	0.210	0.186	0.155	0.071
M10	**−0.308**	**0.016**	0.208	−0.205	0.116	0.177
M10 Onset	−0.242	0.060	0.325	−0.116	0.378	0.518
L5	0.069	0.597	0.809	**−0.266**	**0.040**	0.074
L5 Onset	0.079	0.545	0.807	−0.033	0.802	0.969
IV	−0.045	0.728	0.277	−0.017	0.895	0.686
IS	0.123	0.344	0.819	0.012	0.927	0.685
RA	−0.159	0.222	0.867	0.227	0.081	0.077
CFI	0.015	0.912	0.631	−0.049	0.708	0.910
Wrist temperature						
MESOR	−0.130	0.318	0.299	0.106	0.420	0.269
24 h Amplitude	−0.159	0.220	0.235	0.114	0.387	0.617
Phase	**−0.302**	**0.014 ***	0.104	0.003	0.982	0.788
Sleep characteristics						
Bedtime	0.214	0.098	0.132	−0.161	0.220	0.166
Wake time	−0.013	0.387	0.777	0.137	0.298	0.317
Sleep phase	0.179	0.167	0.353	−0.168	0.201	0.353
Time in bed	−0.078	0.550	0.258	−0.034	0.795	0.657
Total sleep	0.036	0.785	0.571	−0.027	0.840	0.678
Sleep latency	−0.061	0.640	0.850	0.054	0.680	0.773
Sleep efficiency, %	−0.179	0.167	0.336	0.070	0.595	0.618
WASO	**−0.339**	**0.008 ***	**0.049**	0.032	0.220	0.836
Blue light						
MESOR	−0.099	0.446	0.390	−0.187	0.152	0.324
24 h Amplitude	−0.110	0.399	0.344	−0.201	0.123	0.909
Phase	−0.162	0.213	0.668	−0.147	0.262	0.546
M10	−0.082	0.532	0.408	−0.241	0.064	0.186
M10 Onset	−0.130	0.318	0.776	**−0.317**	**0.014**	**0.016**
L5	0.017	0.899	0.085	−0.058	0.659	0.425
L5 Onset	0.182	0.161	0.309	−0.010	0.934	0.679
L5 log_10_	0.178	0.169	0.412	−0.010	0.934	0.995
IV	0.228	0.078	*0.052*	0.215	0.100	0.093
IS	−0.037	0.428	*0.052*	−0.092	0.486	0.571
RA	−0.175	0.177	0.713	−0.079	0.550	0.248
DDI*_bl_*	0.047	0.733	0.762	**0.256**	**0.049**	0.148
NEI*_bl_*	0.155	0.234	0.115	−0.150	0.251	0.650

M10—average value of 10 h of greatest activity or blue light exposure; L5—5 h of lowest activity or blue light exposure; IV—intra-daily variability; IS—inter-daily stability; RA—relative amplitude; DDI*_bl_*—daylight deficiency index of blue light; NEI*_bl—_* nocturnal excess index of blue light; CFI—circadian function index; phases and onset times are indicated in hh:mm; DDI*_bl_* and NEI*_bl_* in μw/cm^2^ × hour; significant differences between the groups, post Tukey HST, and non-zero correlations after FDR correction for multiple testing are in **bold**. * after correction for multiple testing at Benjamini–Hochberg’s FDR = 0.10; MR + CF p—p-value from multiple regression model including all related variables after correction for co-factors of age, sex and population.

**Table 4 biology-13-00022-t004:** Population-, age- and sex-related differences in actigraphy-based indices in Arctic residents during spring equinox (n = 62).

Variable	Population		Age		Sex	
	Non-Natives (n = 38)	Natives (n = 24)	12–16 (n = 14)	20–59 (n = 48)	Males (n = 11)	Females (n = 51)
Activity, PIM						
MESOR	**2263 ± 534**	**3022 ± 706 *****	2803 ± 687	2485 ± 703	2706 ± 823	2525 ± 685
24 h Amplitude	**1596 ± 401**	**2393 ± 695 ****	**2472 ± 748**	**1739 ± 532 ***	2232 ± 969	1834 ± 558
Phase	14:33 ± 1:24	15:05 ± 0:56	14:48 ± 0:49	14:45 ± 1:22	**15:29 ± 1:04**	**14:36 ± 1.15 ***
M10	**3634 ± 812**	**5000 ± 1279 *****	4842 ± 1288	3965 ± 1125	4606 ± 1582	4067 ± 1112
M10 Onset	8:43 ± 1:25	9:27 ± 0:49	9:15 ± 1:00	8:56 ± 1:20	9:32 ± 0:51	8:53 ± 1:19
L5	217 ± 219	222 ± 171	**125 ± 46**	**246 ± 219 ***	151 ± 79	234 ± 215
L5 Onset	1:42 ± 1:20	1:32 ± 1:02	**0:54 ± 0:52**	**1:51 ± 1:14 ***	1:36 ± 1:14	1:39 ± 1:14
IV	0.894 ± 0.173	0.837 ± 0.156	0.881 ± 0.183	0.869 ± 0.165	0.820 ± 0.157	0.883 ± 0.169
IS	0.519 ± 0.098	0.584 ± 0.110	0.597 ± 0.121	0.529 ± 0.098	**0.448 ± 0.122**	**0.565 ± 0.092 ***
RA	0.872 ± 0.073	0.905 ± 0.052	**0.948 ± 0.019**	**0.867 ± 0.065 *****	0.899 ± 0.057	0.882 ± 0.069
CFI	0.651 ± 0.057	0.695 ± 0.054	0.699 ± 0.056	0.660 ± 0.058	0.654 ± 0.065	0.671 ± 0.058
Wrist temperature, °C						
MESOR	**31.63 ± 0.53**	**32.07 ± 0.55 ***	32.26 ± 0.56	31.67 ± 0.51	31.55 ± 0.73	31.86 ± 0.53
24 h Amplitude	**1.49 ± 0.58**	**1.19 ± 0.44 ***	1.29 ± 0.52	1.40 ± 0.55	1.60 ± 0.51	1.33 ± 0.54
Phase	2:25 ± 1:39	3:16 ± 1:18	2:59:1:36	2:41:1:34	3:27 ± 1:02	2:36 ± 1:38
Sleep characteristics, hh:mm						
Bedtime	22:42 ± 1:14	22:42 ± 0:54	22:28 ± 1:01	22:46 ± 1:12	22:58 ± 1:16	22:38 ± 1:05
Wake time	7:01 ± 1:31	6:52 ± 1:09	6:51 ± 2:09	6:59 ± 1:29	7:36 ± 2:06	6:49 ± 1:09
Sleep phase	2:51 ± 1:12	2:45 ± 0:54	2:39 ± 0:45	2:52 ± 1:10	3:17 ± 1:22	2:43 ± 0:57
Time in bed	8:17 ± 1:23	8:08 ± 0:58	8:20 ± 0:59	8:12 ± 1:18	8:38 ± 1:34	8:08 ± 1:09
Total sleep	7:18 ± 1:17	7:17 ± 0:58	7:34 ± 0:54	7:13 ± 1:14	7:50 ± 1:40	7:11 ± 1:01
Sleep latency, min	02.58 ± 2.35	02.21 ± 1.80	1.78 ± 1.58	2.63 ± 2.27	01.92 ± 2.36	02.55 ± 2.12
Sleep efficiency, %	88.67 ± 4.03	85.58 ± 6.24	88.15 ± 6.73	88.15 ± 6.73	88.59 ± 6.35	87.90 ± 4.79
WASO	0:54 ± 0:27	0:47 ± 0:18	0:42 ± 0:17	0:54 ± 0:25	0:45 ± 0:21	0:53 ± 0:24
Blue light, μw/cm^2^						
MESOR	13.00 ± 7.83	8.20 ± 6.36	8.10 ± 6.36	12.03 ± 7.83	**16.98 ± 11.73**	**9.88 ± 5.90 ****
24 h Amplitude	18.61 ± 12.38	12.31 ± 8.30	12.18 ± 10.29	17.34 ± 11.46	**25.21 ± 17.03**	**14.22 ± 8.77 ****
Phase	12:46 ± 0:52	13:07 ± 0:55	**13:27 ± 1:02**	**12:44 ± 0:48 ***	13:30 ± 1:18	12:48 ± 0:46
M10	27.00 ± 16.91	18.00 ± 14.60	18.54 ± 14.60	24.96 ± 15.62	**34.18 ± 21.55**	**21.21 ± 13.06 ***
M10 Onset	7:50 ± 0:46	8:01 ± 0:47	8:10 ± 0:51	7:49 ± 0:44	8:15 ± 0:55	7:50 ± 0:44
L5	0.100 ± 0.169	0.024 ± 0.049	0.003 ± 0.006	0.091 ± 0.154	0.075 ± 0.158	0.070 ± 0.138
L5 Onset	1:34 ± 2:10	1:01 ± 1:01	0:29 ± 0:48	1:36 ± 1:57	1:13 ± 1:21	1:23 ± 1:55
L5 log_10_	−2.02 ± 1.28	−2.60 ± 1.13	**−3.40 ± 0.88**	**−1.90 ± 1.13 ****	−2.35 ± 1.43	−2.22 ± 1.22
IV	0.915 ± 0.341	0.904 ± 0.345	1.030 ± 0.351	0.894 ± 0.334	1.077 ± 0.385	0.892 ± 0.324
IS	0.397 ± 0.153	0.382 ± 0.119	0.362 ± 0.134	0.400 ± 0.142	0.322 ± 0.181	0.406 ± 0.127
RA	0.980 ± 0.029	0.992 ± 0.014	**0.999 ± 0.001**	**0.980 ± 0.026 ***	0.988 ± 0.022	0.984 ± 0.025
DDI*_bl_*	**308 ± 88**	**394 ± 64 ****	**400 ± 96**	**324 ± 119 ***	282 ± 175	354 ± 100
NEI*_bl_*	**2.81 ± 2.39**	**1.12 ± 1.50 *****	**0.89 ± 1.65**	**2.56 ± 2.29 ****	2.55 ± 2.25	2.07 ± 2.28

All values in the groups are mean ± SD. M10—average value of 10 h of greatest activity or blue light exposure; L5—5 h of lowest activity or blue light exposure; IV—intra-daily variability; IS—inter-daily stability; RA—relative amplitude; DDI*_bl_*—daylight deficiency index of blue light; NEI*_bl_*—nocturnal excess index of blue light; CFI—circadian function index; phases and onset times are indicated in hh:mm; DDI*_bl_* and NEI*_bl_* in μw/cm^2^ × hour; significant differences from Mann–Whitney rank-sum test are in **bold:** * *p* < 0.05; ** *p* < 0.01; *** *p* < 0.001.

## Data Availability

The data presented in this study are available upon reasonable request from the corresponding author. The data are not publicly available due to privacy.

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
