# Peer review of "Blue Light and Temperature Actigraphy Measures Predicting Metabolic Health Are Linked to Melatonin Receptor Polymorphism"

_biology, 2023, doi:10.3390/biology13010022_

Round 1

Reviewer 1 Report

Comments and Suggestions for Authors

The authors investigate the relationship between light exposure and metabolic parameters in a human sample of arctic residents. The results provide valuable insight into an understudied demographic which resides in conditions of circadian disruption. In addition the authors introduce a Nocturnal Excess Index which correlated with high BMI, and, following validation, could potentially be used in general population and applied to improve metabolic outcomes.

This research paper is of interest to scientific community and should be published with minor revisions:

General comments:

The authors are requested to standardize the figure legends. For example: some figure legends start with the description of the graphs, while others reference statistics directly.

When referencing figures or tables in the text, they should be referenced in brackets. The referencing could be facilitated by assingning additional reference to individual graphs, for example: A. B. etc.

Line 85: Phrasing of the sense may cause the reader to interpret that ipRGC and mRGC are distinct subpopulations. The reviewer suggest replacing “or” with “also known as” or similar phrasing.

Line 102: Could be misconstrued as a 5 hours delay of least activity onset. The sentence could benefit from clearer phrasing.

Line 191: Since this is the third mention of ActStudio software, I suggest moving the producer information to line 167 (first mention).

Line 236: Figure 1 depicts seasonal exposure patterns. However, I failed to find information on how and when this data was collected or extracted from.

Line 247: Could the authors specify what does “mean values”  average?

Line 330: Description mentions G-allele carriers vs GG genotype – should it not be CC genotype?

Line 337: The reviewer suggest staring with a brief description of data plotted before detailing the statistical analysis.

Line 374: It is not clear to the reviewer what the “# - unless otherwise indicated” means to describe.

Table 4: There is a non-equal distribution of certain traits between groups, for example “natives” as a group are younger than “non natives”. Could the analysis be conducted while controlling for these confounders?

Line 397-399: Please delete the sentences: “This section may be divided by subheadings (…) conclusions that can be drawn.”

Author Response

The authors investigate the relationship between light exposure and metabolic parameters in a human sample of arctic residents. The results provide valuable insight into an understudied demographic which resides in conditions of circadian disruption. In addition the authors introduce a Nocturnal Excess Index which correlated with high BMI, and, following validation, could potentially be used in general population and applied to improve metabolic outcomes.

This research paper is of interest to scientific community and should be published with minor revision.

We are thankful to this reviewer for taking time to evaluate our work and provide useful comments. We addressed each issue point-by-point. All additions and changes in the main text are marked in red.

The authors are requested to standardize the figure legends. For example: some figure legends start with the description of the graphs, while others reference statistics directly.

We now tried to keep all figure legends within the same standard: first sentence – general title. A, B, etc.:, followed by explanations for each particular figure with statistical details when necessary.

When referencing figures or tables in the text, they should be referenced in brackets. The referencing could be facilitated by assingning additional reference to individual graphs, for example: A. B. etc.

We made changes to all figures accordingly.

Line 85: Phrasing of the sense may cause the reader to interpret that ipRGC and mRGC are distinct subpopulations. The reviewer suggest replacing “or” with “also known as” or similar phrasing.

Was corrected using “also known as”

Line 102: Could be misconstrued as a 5 hours delay of least activity onset. The sentence could benefit from clearer phrasing.

Was rephrased as follows: “… – delayed onset of least nocturnal activity and light exposure (as measured by L5 index: 5 hours of least activity/light exposure)...”

Line 191: Since this is the third mention of ActStudio software, I suggest moving the producer information to line 167 (first mention).

As recommended, we now indicated the producer information at line 167, first mention.

Line 236: Figure 1 depicts seasonal exposure patterns. However, I failed to find information on how and when this data was collected or extracted from.

We now added a sentence in the beginning of the Results section as follows: “Participants of “Light Arctic” study provided seven-day actimetry in each season: during winter solstice, spring equinox, summer solstice, and autumn equinox. For this paper, spring was chosen as a reference point to provide seasonal assessments for associations of ambient light environment with circadian physiology, sleep, and metabolism”

Line 247: Could the authors specify what does “mean values”  average?

We now added the following explanation: “Individual mean values of blue light exposure in consecutive 30-min epochs were used to depict the average 24-h patterns of light exposure in each season. Light exposure around the spring equinox was closest to the recommended reference curve for circadian light hygiene”

Line 330: Description mentions G-allele carriers vs GG genotype – should it not be CC genotype?

Thank you for noting this misprint, it was corrected.

Line 337: The reviewer suggest staring with a brief description of data plotted before detailing the statistical analysis.

We now added an explanatory sentence for Fig 3A.B: “Lower wT MESOR associates with BMI in G-allele carriers, but not in CC genotype.”, and rephrased legend to Fig 3 C,D: “24-h patterns show lower wT, particularly in the morning, in overweight G-allele carriers than in normal-weight G-allele carriers (C), this difference is absent in CC genotype (D).

Line 374: It is not clear to the reviewer what the “# - unless otherwise indicated” means to describe.

Thank you for noting. We removed this sentence and units specifications legend of Table 3.

Table 4: There is a non-equal distribution of certain traits between groups, for example “natives” as a group are younger than “non natives”. Could the analysis be conducted while controlling for these confounders?

Thank you for this comment. We applied corrections for confounding factors such as population, age, and sex to all regression analyses, as shown in Tables 2-3. The purpose of Table 4 was to show the raw data of these different age groups.

Line 397-399: Please delete the sentences: “This section may be divided by subheadings (…) conclusions that can be drawn.”

Thank you for noting. We removed this sentence; it was part of the journal's template form.

Reviewer 2 Report

Comments and Suggestions for Authors

The authors approached a subject of interest, analyzing the effect of melatonin receptor polymorphism on circadian rhythm and metabolic risks. The article is extensively documented and well-written. However, there are some areas for improvement as follows: 

1. I suggest restructuring the abstract to provide background information, objectives, methodology, results, and conclusion, which would enhance its clarity and organization. 

2. My recommendation is to provide more information about the rs10830963 gene polymorphism, including relevant literature information about the gene and its roles supported by references. Currently, it is only briefly mentioned in the introduction section.

3. The authors should have outlined the study objectives more clearly. The result section should be structured according to these objectives.

4. I suggest labeling multiple images in figures with letters (e.g. Figure 1A, 1B) instead of using directional descriptions (e.g. lower row left).

Comments on the Quality of English Language

Some phrases are too long and hard to follow. Try to separate them into shorter and clearer sentences to increase readability.

Author Response

The authors approached a subject of interest, analyzing the effect of melatonin receptor polymorphism on circadian rhythm and metabolic risks. The article is extensively documented and well-written. However, there are some areas for improvement as follows: 

We are thankful to this reviewer for taking time to evaluate our work and provide useful comments. We addressed each issue point-by-point. All additions and changes in the main text are marked in red.

1. I suggest restructuring the abstract to provide background information, objectives, methodology, results, and conclusion, which would enhance its clarity and organization. 

As recommended we restructured the abstract as follows: “This study explores the relationship between light features of Arctic spring equinox and circadian rhythms, sleep, and metabolic health. Residents (N=62) provided week-long actigraphy measures, including light exposure, which were related to body mass index (BMI), leptin and cortisol. Lower wrist temperature (wT) and higher evening blue light exposure (BLE), expressed as a novel index, the Nocturnal Excess Index (NEIbl), were the most sensitive actigraphy measures associated with BMI. A higher BMI was linked to nocturnal BLE within distinct time windows. These associations were present specifically in carriers of the MTNR1B rs10830963 G-allele. A larger wake-after-sleep onset (WASO), smaller 24-hour amplitude and earlier phase of the activity rhythm were associated with higher leptin. A higher cortisol was associated with an earlier M10 onset of BLE and with our other novel index, the Daylight Deficit Index of blue light, DDIbl. We also found sex-, age-, and population-dependent differences in parametric and non-parametric indices of BLE, wT and physical activity, while there were no differences in any sleep characteristics. Overall, this study determined sensitive actigraphy markers of light exposure and wT predictive of metabolic health and showed that these markers are linked to melatonin receptor polymorphism.

2. My recommendation is to provide more information about the rs10830963 gene polymorphism, including relevant literature information about the gene and its roles supported by references. Currently, it is only briefly mentioned in the introduction section.

We thank the reviewer for this valuable suggestion. We now added an additional information into the paragraph discussing the rationale to include this particular snp into analyses: “The pineal gland hormone melatonin is a well-documented substance that mediates the influence of outdoor light cues on the inner circadian machinery. Seasonal changes in the length of the dark phase lead to parallel shifts in the duration and timing of the onset of melatonin secretion. Melatonin exerts its effects via its own receptors, MTNR1A and MTNR1B, as well as through receptor-independent mechanisms [15, 108]. Polymorphic variations in the MTNR1B gene have a pofound effect on gene expression and protein properties. Associations between carriage of the MTNR1B rs10830963 minor G-allele with the affected modulation of the metabolic response to melatonin and light was evident in previous studies [42–44]. Carriers of the minor G-allele of the MTNR1B rs10830963 snp r a large part of populations, which have elevated fasting glucose and higher risks of metabolic disorders, particularly type 2 diabetes mellitus [41]. They also have changed and prolonged patterns of melatonin secretion [42] that can be related to chronotype [43], or reduced light signaling due to retinal ganglion cells loss [44].”.

3. The authors should have outlined the study objectives more clearly. The result section should be structured according to these objectives.

As also suggested by another reviewer, we added general information in section 1 of the results, providing an explanation on the comparison of data between seasons. We also underlined the main study objectives in the abstract, as follows: ““This study explores the relationship between light features of Arctic spring equinox with circadian rhythms, sleep, and metabolic health.” We believe that the current structure of Results follows the study objectives consecutively: In 3.1. “Seasonal features of light exposure in the Arctic. Spring as a season with the most favorable circadian light environment” we explained we started our presentation with spring equinox, as “reference” season with least compromised circadian light exposure in the Arctic. Next we provided “3.2 General characteristics of participants of the spring equinox session”. In 3.3. Blue light nocturnal excess and lower wrist temperature predict Body Mass Index we introduced most sensitive measures that were associated with BMI. In 3.4 “MTNR1B polymorphism accounts for the interaction between light, wrist temperature and metabolic health” we emphasized that measures introduced in 3.3 relate to certain gene polymorphism. In 3.5. (“Associations of leptin and cortisol with actimetry-derived indices”) we referred to proxies of metabolic health that were linked to different actigraphic measures from that of BMI. Finally, in 3.6, we provided additional information on raw data of co-factors of interest “Age-, sex- and indigeneity-related aspects of actigraphy-based indices”, that we used in correlation models.

4. I suggest labeling multiple images in figures with letters (e.g. Figure 1A, 1B) instead of using directional descriptions (e.g. lower row left).

As also suggested by another reviewer, we now labeled all figures according to this recommendation.

Comments on the Quality of English Language/

Some phrases are too long and hard to follow. Try to separate them into shorter and clearer sentences to increase readability.

We tried to separate several longer sentences in the Introduction and Discussion into shorter ones. These sentences as marked in red in the main text. We also rearranged sentences in figure legends as recommended by another reviewer.